# Genetic variation for fusarium crown rot tolerance in durum wheat

**Gururaj Pralhad Kadkol**[1]*, **Jess Meza**[2], **Steven Simpfendorfer**[1], **Steve Harden**[1], **Brian Cullis**[2]

**1** Tamworth Agricultural Institute, NSW DPI, Calala, New South Wales, Australia, **2** Centre for Bioinformatics and Biometrics, National Institute for Applied Statistics Research Australia, University of Wollongong, Wollongong, New South Wales, Australia

\* Gururaj.Kadkol@dpi.nsw.gov.au

**Data Availability Statement:** All data underlying the results published in this study are available within the Supporting information files and also from the first author, Gururaj Kadkol (gururaj.kadkol@dpi.nsw.gov.au).

## Abstract

Tolerance to the cereal disease Fusarium crown rot (FCR) was investigated in a set of 34 durum wheat genotypes, with Suntop, (bread wheat) and EGA Bellaroi (durum) as tolerant and intolerant controls, in a series of replicated field trials over four years with inoculated (FCR-i) and non-inoculated (FCR-n) plots of the genotypes. The genotypes included conventional durum lines and lines derived from crossing durum with 2–49, a bread wheat genotype with the highest level of partial resistance to FCR. A split plot trial design was chosen to optimize the efficiency for the prediction of FCR tolerance for each genotype. A multi-environment trial (MET) analysis was undertaken which indicated that there was good repeatability of FCR tolerance across years. Based on an FCR tolerance index, Suntop was the most tolerant genotype and EGA Bellaroi was very intolerant, but some durum wheats had FCR tolerance indices which were comparable to Suntop. These included some conventional durum genotypes, V101030, TD1702, V11TD013*3X-63 and DBA Bindaroi, as well as genotypes from crosses with 2–49 (V114916 and V114942). The correlation between FCR tolerance and FCR-n yield predictions was moderately negative indicating it could be somewhat difficult to develop FCR-tolerant genotypes that are high yielding under low disease pressure. However, FCR tolerance showed a positive correlation with FCR-i yield predictions in seasons of high disease expression indicating it could be possible to screen for FCR tolerance using only FCR-i treatments. These results are the first demonstration of genetic diversity in durum germplasm for FCR tolerance and they provide a basis for breeding for this trait.

## Introduction

Fusarium crown rot (FCR), caused by the fungus *Fusarium pseudograminearum* (*Fp*), is an important disease of cereals in Australia and other countries, such as, USA, South Africa, North Africa, Italy, Middle East and China [1–4]. It is the most important disease in durum wheat (*Triticum durum* desf.) production in northern New South Wales (NSW) and southern Queensland and it occurs in all cereal growing regions of Australia [5, 6]. *Fp* infects the crown

**Funding:** This research was conducted under the Durum Breeding Australia project (GK, No. 9175799) funded by the GRDC (https://access.grdc.com.au/), NSW DPI (http://www.dpi.nsw.gov.au/) and The University of Adelaide (https://www.adelaide.edu.au/). The funders had no role in study design, data collection and analysis, decision to publish or preparation of the manuscript.

**Competing interests:** The authors have declared that no competing interests exist.

and tiller bases, causing a characteristic brown discolouration with the severity of this symptom used to visually assess the relative resistance of cereal varieties [7]. However, yield loss associated with FCR infection is related to the expression of whiteheads during grain-filling [2]. *Fp* has a wide range of winter cereal and grass hosts and survives as fungal hyphae in the residues of infected plants for extended periods [8]. Therefore, the main management option for the control of FCR is practicing good crop rotation sequences involving non-cereal crops such as canola and grain legumes [9, 10]. Currently, there are no effective seed or foliar fungicide products available for management of FCR [1]. Climate change could increase the frequency of drought conditions where increased moisture and temperature stress occurs during grain-filling. Such conditions are known to exacerbate the severity of FCR infection and expression as whiteheads [11–13]. Furthermore, adoption of conservation agriculture practices is essential for adapting to climate change, but these practices also promote FCR development through stubble retention [4]. It is therefore important to develop genetic resistance and tolerance for FCR in durum to reduce loss of grain yield from this disease. In Australia, the viability of the durum industry is considered to depend upon being able to improve FCR resistance and tolerance of durum wheat to levels comparable to those found in the current bread wheat varieties.

Tolerance, as a trait, relates directly to the effect of the disease on grain yield and therefore is generally a more useful and practically relevant trait than resistance for the profitability of durum growers. Resistant varieties also need to be tolerant to the disease to reduce damage to the crop under heavy infestation [14]. Tolerance to a disease is defined as the ability of a host to limit the damage or impact of a given pathogen burden on host health [15]. Thus, FCR tolerant cultivars would lose less yield or maintain levels of grain production in the presence of FCR infection, compared with other cultivars. Resistance involves a mutual incompatibility between the host and the pathogen allowing the host to prevent or limit the growth of the pathogen [15]. Thus, resistance and tolerance are separate and often unrelated traits [14, 15] and the expression of resistance is different from that of tolerance.

Tolerance is assessed experimentally by comparing the yield performance of genotypes under high disease pressure and low or nil disease pressure. With FCR the differential disease pressure is achieved in field trials by varying the amount of inoculum to which plants are exposed. There are two main inoculation methods, being either inoculating genotype seed with spores [13], or, delivering the inoculum to the furrow on sterilised grain with the seed of genotypes during sowing [16–19].

There have been varied approaches to estimating genotype tolerance to diseases. The main method is to include non-inoculated, low disease or disease-free control plots [14, 17, 19, 20]. There are also studies that have not included non-inoculated or disease-free control plots but have used a single high level of disease pressure, thus attributing yield differences between cultivars to differences in tolerance [14, 21]. Whilst including non-inoculated or disease-free control plots makes it possible to differentiate between inherent differences in yield potential and tolerance to the disease, their omission may be practical for preliminary screening trials. Tolerance has often been calculated as the difference or ratio between yields measured from inoculated and non-inoculated field plots [7, 19]. To accurately attribute differences in yield to greater disease pressure, assessments of the pathogen burden need to be undertaken in all plots to avoid misleading estimates of tolerance, since non-inoculated plots could have disease present [17].

FCR screening and evaluation, until recently, has focussed solely on resistance and is largely based on evaluating basal browning symptoms of seedlings after growing through a layer of *Fp* inoculum in soil in glasshouse pot trials [7]. A high positive correlation between seedling symptoms from glasshouse trials and field symptoms have been established, and hence

breeding programmes have widely used seedling screening as a predictor of FCR resistance [6]. Quantitative trait loci (QTLs) associated with FCR resistance have been identified in bread wheat using this glasshouse-based phenotypic assessment [7, 22–24]. Other glasshouse screening methods have been developed based on different procedures for inoculation [24–26]. An outdoor pot assay known as the "terrace" system involving growing plants on terraces in open ended tubes containing 0.24 g of FCR inoculum has been routinely used for screening bread wheat genotypes in South Australia [27], but this method suffers from high variability [7]. FCR screening in field-based disease nurseries is also commonly practised (for example, [28]) wherein sterilised durum grain colonised with *Fp* is added to the furrow at the rate of 2 g/m of row. However, there is an emerging trend to focus on FCR tolerance in bread wheat pre-breeding, demonstrating the value of tolerance over resistance [17, 20, 29].

Despite the importance of FCR in durum wheat, there is very little published information on targeted assessment of genetic variation for FCR tolerance or resistance in durum germplasm [7]. There are no previous reports of FCR tolerance, but a small number of studies have reported on an absence of variation for FCR resistance in durum germplasm. Wallwork et al. [27] tested a set of 90 *T. dicoccum* genotypes and an unspecified number of durum cultivars from a variety of sources, using the "terrace" system, and found partial resistance in four *T. dicoccum* genotypes. Ma et al. [30] reported absence of variation for resistance in 400 unspecified durum genotypes using a glasshouse test. In many studies the assessment of resistance has been based on seedling studies which might have been able to identify only the best resistance but not the intermediate or partial resistance which might be expressed in adult plants [27]. Also, all the screening and pre-breeding efforts in Australia [e.g. 28], to date, have focussed on symptom-based assessment of resistance to FCR but the correlation of this resistance with yield or yield loss is uncertain. The ability of durum genotypes developed from pre-breeding research with improved resistance to FCR to maintain or improve production levels in the presence of this disease has yet to be determined.

The aim of this research was to investigate genetic variation for FCR tolerance with an emphasis on current elite Australian durum breeding material in the DBA program. Previous studies of FCR tolerance have used several levels of disease pressure to determine tolerance in intensive tests of a small number of genotypes [17, 31]. However, this approach is not suitable for estimating genetic variation in a breeding setting because a higher number of genotypes need to be tested without making the experiments too large and cost prohibitive. Therefore, we considered it adequate to use one standard level of disease pressure along with non-inoculated control plots to develop a practical testing method for FCR tolerance within a breeding context.

## Materials and methods

### Field trials

The trials were conducted at Tamworth Agricultural Institute between 2015 and 2017. In 2018 the trial was moved to the Liverpool Plains Field Station, Breeza, and conducted under irrigation due to drought conditions and lack of soil moisture at planting in Tamworth. Initial trials were conducted in previous years to determine the most reliable trial protocols, including the most appropriate trial design and sowing date because FCR data tends to be highly variable (data not included). Each field trial had four replications. Plots were 2 m wide and 10 m long. Within a trial, genotypes were grown as both inoculated with FCR (FCR-i), and as non-inoculated bare seed (FCR-n) side-by-side using a split plot design with genotypes as main plots and FCR treatments allocated randomly to the subplots. Rainfall and temperature details for 2015–2018 are summarised in Fig 1.

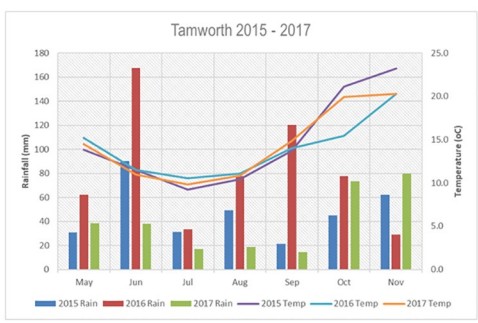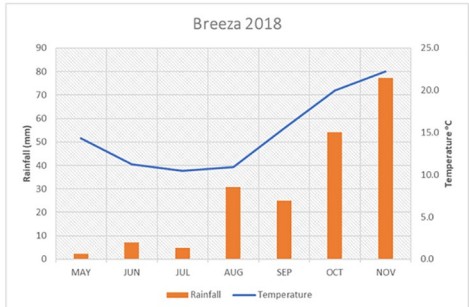

**Fig 1. Monthly rainfall (primary Y axis) and average temperature (secondary Y axis) summary for Tamworth and Breeza trial sites.**

PREDICTA® B DNA tests [32] were conducted on soil samples collected prior to sowing from the trial area to determine the background concentration of *Fp* along with levels of a range of other soil-borne pathogens. Plots inoculated with FCR were sown with inoculum mixed with viable seed at a rate of 2 g *Fp* inoculum/m row, as described by Dodman and Wildermuth [16]. Details of the trial sites, sowing dates and agronomic management are outlined in Table 1. Plots were harvested using a Kingaroy Engineering Works plot harvester at maturity to determine grain yield.

## Germplasm

The study comprised of a set of 34 durum genotypes (Table 2). Most of the genotypes were from Durum Breeding Australia (DBA, a joint project between New South Wales Department of Primary Industry (NSW DPI), The University of Adelaide and the Grains Research and Development Corporation). Additionally, a tolerant bread wheat control, Suntop, and an intolerant durum control, EGA Bellaroi, were included. The genotypes were selected for their promising response to FCR observed in previous breeding and pre-breeding trials. This set also included genotypes derived from interspecific crosses between a FCR resistant bread wheat genotype, 2–49, and advanced DBA durum genotypes that were developed in a joint FCR pre-breeding project at NSW DPI and the University of Southern Queensland conducted from 2004–2009 [28]. Some genotypes judged to be FCR-intolerant were replaced with new genotypes that were selected based on their performance in other FCR treated trials in the DBA North breeding program to improve the chances of identifying FCR-tolerant genotypes.

**Table 1. Details of trial sites and agronomic management.**

|  | Year | | | |
|---|---|---|---|---|
|  | **2015** | **2016** | **2017** | **2018** |
| Location | Tamworth | Tamworth | Tamworth | Breeza |
| Latitude | 31.09˚S | 31.09˚S | 31.09˚S | 31.25˚S |
| Longitude | 150.93˚E | 150.93˚E | 150.93˚E | 150.46˚E |
| Altitude (m) | 404 | 404 | 404 | 295 |
| Soil Classification | Grey cracking clay | Grey cracking clay | Grey cracking clay | Grey cracking clay |
| pH (CaCl2) | 7.4 | 6.1 | 6.1 | 7.9 |
| PREDICTA® B | FCR–BDL | FCR–BDL, *Pt*–low | FCR–BDL, *Pt*–medium | FCR–BDL, *Pt*–medium, *Bp*–low |
| Sowing dates | 22/07/2015 | 17/06/2016 | 16/06/2017 | 26/06/2018 |

BDL = below detectable limits, *Pt* = *Pratylenchus thornii*, *Bp* = *Bipolaris*

**Table 2. Genotypes tested in this study and their details.**

| GENOTYPE NAME | PEDIGREE | STATUS | BREEDER | YEARS TESTED |
|---|---|---|---|---|
| ZDBO4-17 | RASCON_21/3/MQUE/ALO//FOJA, CDSS94Y00099S-7M-0Y-0B-1Y-0B-0BLR-3Y-0B | Durum breeding line | CIMMYT | 2015–17 |
| V100952 | 230349/260233 | Durum breeding line | NDBA | 2015–16 |
| V101030 | JANDAROI/200856. | Durum breeding line | NDBA | 2015–18 |
| V240578 | 960707/980947 | Durum breeding line | NDBA | 2015–17 |
| V280545 | 200856/980990 | Durum breeding line | NDBA | 2015–17 |
| V280617 | 200419/980012 | Durum breeding line | NDBA | 2015–16 |
| V280973 | 200856/980990 | Durum breeding line | NDBA | 2015–16 |
| V281019 | 980012/200777 | Durum breeding line | NDBA | 2015 |
| V290222 | 230800/234193 | Durum breeding line | NDBA | 2015 |
| V290328 | 230800/980019 | Durum breeding line | NDBA | 2015 |
| V290491 | 230616/230800 | Durum breeding line | NDBA | 2015–17 |
| V290564 | 230616/230800 | Durum breeding line | NDBA | 2015–17 |
| TD1601 | 230726/SUNVALE | Durum breeding line | NDBA | 2016–18 |
| TD1602 | 234194/YAWA | Durum breeding line | NDBA | 2016–18 |
| TD1701 | 234194/YAWA | Durum breeding line | NDBA | 2017–18 |
| TD1702 | CAPAROI/WID002 | Durum breeding line | NDBA | 2017–18 |
| V114906 | 2-49/EGABELLAROI (= 2-49A 9–5) | *T. aestivum X T. durum* | NSWDPI | 2015–17 |
| V114908 | 2-49/EGABELLAROI (= 2/49 A 18–6) | *T. aestivum X T. durum* | NSWDPI | 2015 |
| V114916 | 2-49/EGABELLAROI (= 2/49A30–5) | *T. aestivum X T. durum* | NSWDPI | 2015–18 |
| V114926 | 2-49/950329 (= 2/49 B 1–6) | *T. aestivum X T. durum* | NSWDPI | 2015–17 |
| V114928 | 2-49/950329 (= 2/49 B 1–6) | *T. aestivum X T. durum* | NSWDPI | 2015–18 |
| V114932 | 2-49/950329 (= 2/49 B 22–2) | *T. aestivum X T. durum* | NSWDPI | 2015–17 |
| V114942 | 2-49/950329 (= 2/49 B 31–10) | *T. aestivum X T. durum* | NSWDPI | 2015–18 |
| V10TD033*3X-098 | DBALILLAROI/HYPERNO | Durum breeding line | NSWDPI | 2018 |
| V11TD013*3X-63 | WID096/DBALILLAROI | Durum breeding line | NSWDPI | 2018 |
| HYPERNO | KALKA 'S'/TAMAROI | Released durum | AGT | 2015–18 |
| EGA BELLAROI | 920405/920274 | Released durum | NSWDPI | 2015–18 |

(*Continued*)

**Table 2.** (Continued)

| GENOTYPE NAME | PEDIGREE | STATUS | BREEDER | YEARS TESTED |
|---|---|---|---|---|
| DBA BINDAROI | CAPAROI/261102 | Released durum | NDBA | 2016–18 |
| DBA LILLAROI | 960273/980596 | Released durum | NDBA | 2015–18 |
| DBA VITTAROI | 200856/980990 | Released durum | NDBA | 2016–18 |
| CAPAROI | LY2.6.3/ 930054 | Released durum | NSWDPI | 2015–18 |
| JANDAROI | (SOURI/WOLLAROI)/KRONOS | Released durum | NSWDPI | 2015–18 |
| DBA AURORA | TAMAROI*2/KALKA//RH920318/KALKA/3/KALKA*2/TAMAROI | Released durum | SDBA | 2015–18 |
| YAWA | ((WESTONIA/KALKA derivative)//(KALKA/TAMAROI))/// (RAC875/KALKA)// TAMAROI)) | Released durum | SDBA | 2015–18 |
| TJILKURI | BRINDUR/3/YALLAROI*2//DURA/YALLAROI/4/RAC875/3/LINGZHI/YALLAROI// TAMAROI/5/LINGZHI/YALLAROI// TAMAROI/3/LINGZHI/YALLAROI | Released durum | SDBA | 2015–17 |
| SUNTOP | ('SUNCO'/2*'PASTOR')/SUN436E | Released bread wheat | AGT | 2015–18 |

CIMMYT–International Centre for Maize and Wheat Improvement, NDBA–DBA North breeding program, SDBA–DBA South breeding program, NSW DPI–New South Wales Department of Primary Industry, AGT = Australian Grain Technology

However, a fairly high degree of concurrence between trials was maintained allowed the MET analysis of the data as discussed below.

### Fusarium crown rot inoculum

Durum grain was sterilised twice at 121˚C for 60 min on two consecutive days then inoculated with a macroconidial suspension of *Fp* prepared in mung bean broth (40 g mung beans in 1 L H₂O, boiled for 30 min, strained and autoclaved at 121˚C for 20 min). Inoculum consisted of an equal mixture of five separate batches of durum grain each of them inoculated with a different aggressive isolate of *Fp* (Table). Each isolate was grown through the non-viable durum grain for three weeks at 25˚C then air dried at 30˚C before use [17].

Although FCR isolates have been shown to vary in aggressiveness, there is no race structure established that causes differential varietal reactions [33]. The details of the five isolates used in preparation of the inoculum for this study (Table 3) were established from hyphal tip cultures to ensure purity and identified by qPCR [32] to be *Fusarium pseudograminearum* by South Australian Research and Development Institute (SARDI). Isolates were collected from commercial crops in several locations in northern NSW with severe basal browning characteristic of Fusarium crown rot infection at harvest in either 2013 or 2014 (Table 3).

### Assessment of FCR infection

FCR symptoms based on the incidence and severity of browning of infected tillers were visually assessed for 25 plants randomly sampled from the middle three rows of each five-row plot

**Table 3. Details of *Fp* isolates used in preparation of CR inoculum.**

| SARDI ID | Isolate ID | Year | State | Location | Host | qPCR result |
|---|---|---|---|---|---|---|
| 4093 AC29312 | CAS-13/94C | 2013 | NSW | Walgett | Wheat | *F. pseudograminearum* |
| 4093 AC29363 | CAS-13/131C | 2013 | NSW | Rowena | Wheat | *F. pseudograminearum* |
| 4095 AC29475 | CAS-13/161N | 2013 | NSW | Moree | Durum | *F. pseudograminearum* |
| 4849 BA48166 | CAS-14/98C | 2014 | NSW | Warren | Wheat | *F. pseudograminearum* |
| 4719 BA40558 | CAS 14/88N | 2014 | Qld | Moonie | Wheat | *F. pseudograminearum* |

in the experiment in the 2016 and 2017 seasons. These plants were collected after harvest, carefully, to preserve the sub-crown internode where possible. For each plant the total number of tillers, along with the number of tillers that exhibited basal browning symptoms, were recorded. By determining the number of tillers that exhibited basal browning from the total number of tillers, a measure of FCR incidence was derived for each plant. The extent of browning was scored between 0 and 3, in 0.5 increments for each tiller and averaged across all tillers on each plant, to provide a measure of FCR severity for each plant, where

- 0 = no browning,

- 0.5 = partial browning 0–2 cm,

- 1 = complete browning 0–2 cm,

- 1.5 = complete browning 0–2 cm + partial browning 2–4 cm,

- 2 = complete browning 0–4 cm,

- 2.5 = complete browning 0–4 cm + partial browning 4–6 cm,

- 3 = complete browning 0–6 cm.

A crown rot index (CRI) was calculated for each plant, using the equation below:

$$CRI = \frac{Tillers\ with\ Basal\ Browning}{Total\ Number\ of\ Tillers} \times \frac{Extent\ of\ Basal\ Browning}{3} \times 100$$

The resulting CRI values ranged from 0 if no tillers on a plant displayed basal browning to 100 if all tillers on a plant displayed basal browning and this browning was complete from 0 to 6 cm [17].

## Statistical methods

Here the analysis of FCR resistance and FCR tolerance is described, which involved fitting an appropriate linear mixed model (LMM) which was commensurate with the aims of the experiment and the structure of the data set. An extended split-plot LMM was used for the analysis of CRI for each of two seasons (viz, 2016 and 2017), whereas a factor analytic LMM [34] was used for the analysis of grain yield for the multi-environment data set spanning 2015–2018. All LMMs were fitted using ASReml-R [35], which provided residual maximum likelihood (REML) estimates of variance parameters and empirical best linear unbiassed predictions (E-BLUPs) of random effects.

## Preamble

The trial designs were all split-plot designs with four blocks. The treatment factors were genotypes and FCR *treatments*. Note that the use of the word "treatment" in describing the FCR *treatment* is not to be confused with the reserved statistical term for the definition of the entire description of what was applied to an experimental unit [36]. Hence the use of the italicised font to make the distinction clear. Words for factor names which were used in the statistical modelling scripts are in "Courier New" font. The two levels of FCR *treatment* were FCR inoculated and FCR non-inoculated, and these will be referred to by FCR-i and FCR-n respectively. The genotypes were allocated to the mainplots and the FCR *treatments* were allocated to the subplots. Plot factors were defined as block, mainplot and subplot with 4, *m* and 2 levels respectively where *m* was the number of genotypes used in the trial. The coded FCR *treatment* factor was shortened to `FCRTrt` for brevity. The term environment was synonymous with

trial and year, and for consistency with the literature on multi-environment trial data sets the factor name environment (or a shortened version Env) was used in the following. Lastly, FCR tolerance was defined as the yield difference between the FCR-i and FCR-n *treatments*.

## Analysis of CRI

The baseline or standard LMM which reproduces the classical analysis (of variance) of a split-plot design includes terms in the fixed effects for the main effects and interactions of the main-plot and subplot treatment factors. In this example this required fitting of the main effects of Genotype and FCRTrt and their interaction (Genotype:FCRTrt), as well as two additional terms fitted as random effects which were denoted by Block and Block:Mainplot. The inclusion of these random terms ensured the LMM reproduced the strata for a split plot analysis. The LMM was extended to account for additional non-treatment sources of variation (should these exist) and the terms Genotype and Genotype:FCR were fitted as random terms which was commensurate with the aim of identifying sources of genetic resistance to FCR in durum. The latter extension of random effects for treatment factors led to a compound symmetric (CS) variance model for the nested effects of genotypes within FCR *treatments*, hence an extension of the CS model which introduced an additional variance parameter to account for possible variance heterogeneity between the two levels of FCR *treatments* was also considered. This model was referred to as the CORGH model. The fit of the CS and CORGH variance models was assessed using AIC values [37]. Formal test of the strength of the agreement between the effects of FCR-i and FCR-n within genotypes, along with testing the presence of genetic variance within each of the two levels of FCR *treatment*, were undertaken using a REML likelihood ratio test statistic [38] applied to appropriate nested variance models.

## Analysis of grain yield

In this section the term of most interest was the compound (random) term which corresponds to the Genotype by FCRTrt by Env (VFE) effects. These terms were assumed to be ordered as genotypes within FCR treatments within environments and the associated variance matrix for the VFE effects were denoted as $\boldsymbol{G}$. Variance parameters and effects associated with this term will be referred to as genetic variance parameters and genetic effects.

**Formulating the baseline model.** In standard multi-environment (MET) data sets which have a simple treatment structure, usually genotypes, formulation of the baseline model commences with fitting a model that assumes independence of the genotype by environment (VE) effects between environments. This model, termed the DIAGONAL variance model for the VE effects is analogous to analysing each environment separately. This baseline model is used to assess whether additional terms are required to account for non-treatment sources of variation as well as investigating the presence of outlier observations. For METs with a factorial treatment structure, such as the factorial combination of FCR *treatments* and genotypes, it is generally preferable to incorporate this structure into the baseline model. Hence the main effects of FCRTrt and Env and the interaction Env:FCRTrt, were fitted as fixed effects and a variance model for the nested effects of FCR treatment by genotype within environment and FCR *treatments* was chosen given by

$$\boldsymbol{G} = \oplus_{i=1}^{4}(\boldsymbol{G}_{2i} \otimes \boldsymbol{I}_{36})$$

where each $G_{2i}$ is a CORGH form but with different variance parameters for each of the four environments. This is analogous to analysing each environment separately (and with the factorial treatment structure of FCR *treatment* and genotype). The ASReml-R call for the baseline model is presented in the S1 File.

Additional random model terms were fitted to accommodate non-treatment sources of variation, as well as random model terms which are associated with the plot structure for a split-plot design. Variance models for these random model terms allowed for variance heterogeneity between environments. The variance model for the residuals was either a one dimensional first order autoregressive variance model or a two dimensional separable first order autoregressive variance model [39], with different variance parameters for each environment. This LMM was referred to as the baseline model.

A formal test for the presence of genetic variance related to FCR tolerance was conducted by fitting a variance model in which the correlation between the effects of FCR-i and FCR-n for each genotype was fixed at unity. This variance model was equivalent to testing that the interaction between FCR *treatment* and genotype was zero and it could be fitted using the fully reduced models proposed by Thompson et al. [40].

**Factor analytic variance models for the VFE effects.** After fitting the baseline model, variance models which allowed for a correlation between the VFE effects in different environments were fitted. To examine the best fitting variance model a separable variance model was considered which was similar to that proposed by Smith et al. [34] and a non-separable model (for the environment and FCR *treatments*). The separable variance model was given by

$$\boldsymbol{G} = \boldsymbol{G}_1 \otimes \boldsymbol{G}_2 \otimes \boldsymbol{G}_3$$

where each $\boldsymbol{G}_i$, $i = 1, 2, 3$ represented a scaled variance matrix associated with each of the three components. That is $\boldsymbol{G}_1$ was a symmetric positive definite matrix of size 4 which was associated with environments, $\boldsymbol{G}_2$ was a symmetric positive definite matrix of size 2 which was associated with FCR *treatments* and $\boldsymbol{G}_3 = \boldsymbol{I}_{36}$ is an identity matrix of size 36 associated with genotypes, 36 being the total number of genotypes tested across the four environments. This model was parsimonious and typically easy to fit but it may not have been appropriate, as it assumes, for example, that the correlation between the FCR-i and FCR-n effects within each genotype is the same for each of the four environments.

The $\boldsymbol{G}_i$, $i = 1, 2, 3$, were assumed to have a factor analytic structure of order 1 (denoted as FA(1)), a CORGH structure and an identity matrix for the environments, FCR treatments and genotypes dimensions respectively. A FA(1) variance model was developed from a latent regression model for the set of VFE effects and was given by

$$\boldsymbol{u} = (\boldsymbol{\Lambda}_s \otimes I_2 \otimes I_{36})\boldsymbol{f}_s + \boldsymbol{\delta}_s$$

where $\boldsymbol{u}$ is the $8m \times 1$ vector of VFE effects, $\boldsymbol{\Lambda}_s$ is a $4 \times 1$ matrix of environment loadings, $\boldsymbol{f}_s$ is the $2m \times 1$ vector of scores for the combinations of FCR *treatment* and genotype, and $\boldsymbol{\delta}_s$ is a set of lack of fit effects or deviations from the latent regression model. From this latent regression model and based on the assumed forms for the variance models for $\boldsymbol{f}_s$ and $\boldsymbol{\delta}_s$ (see Smith et al. (2019)) then it followed that the variance for the set of VFE effects was given by

$$\text{var}(\boldsymbol{u}) = (\boldsymbol{\Lambda}_s \boldsymbol{\Lambda}_s^T + \boldsymbol{\Psi}_s) \otimes \boldsymbol{G}_2 \otimes \boldsymbol{I}_{36}$$

where $\boldsymbol{\Psi}_s$ is a $4 \times 4$ diagonal matrix whose non-zero elements are the specific variances for each environment. One of the variance parameters in $\boldsymbol{G}_2$ is set to one to ensure identifiability of the variance model.

The non-separable variance model required definition of a factor, called EnvFCRTrt with 8 levels which were the combinations of the four environments and the two levels of the FCR *treatment*. The variance matrix for the VFE effects was

$$\boldsymbol{G} = \boldsymbol{G}_{ns} \otimes \boldsymbol{I}_{36}$$

where $G_{ns}$ was an $8 \times 8$ symmetric matrix whose columns and rows were indexed by the levels of `EnvFCRTrt` and was assumed to have a FA($k$) structure. The latent regression model for the set of VFE effects was

$$\boldsymbol{u} = (\boldsymbol{\Lambda}_{\text{ns}} \otimes I_{36})\boldsymbol{f}_{ns} + \boldsymbol{\delta}_{ns}$$

where $\boldsymbol{\Lambda}_{\text{ns}}$ is a $8 \times k$ matrix of environment by FCR *treatment* loadings, where $k$ could be 1, 2 or 3 since there were eight levels in the `EnvFCRTrt`, $\boldsymbol{f}_{ns}$ is the $m \times 1$ vector of scores for genotypes, and $\boldsymbol{\delta}_{ns}$ is a set of lack of fit effects or deviations from the latent regression model. Using the latent regression model and standard assumptions regarding the variance models for $\boldsymbol{f}_{ns}$ and $\boldsymbol{\delta}_{ns}$ led to the following variance model for the set of VFE effects:

$$\text{var}(\boldsymbol{u}) = (\boldsymbol{\Lambda}_{ns}\boldsymbol{\Lambda}_{ns}^T + \boldsymbol{\Psi}_{ns}) \otimes \boldsymbol{I}_{36}$$

where $\boldsymbol{\Psi}_s$ is a $8 \times 8$ diagonal matrix whose non zero elements are the specific variances for each level of `EnvFCRTrt`.

**Constructing an FCR tolerance index.**   The presence of genetic variance for tolerance was summarised by constructing a FCR tolerance index which is based on the yield difference of a genotype between the FCR-n and FCR-i *treatments*, but exploits the underlying form of the FA structure, in particular its analogy with multiple linear regression following Smith and Cullis [41]. Smith and Cullis [41] developed factor analytic selection tools (FAST) which include natural measures of overall (yield) performance, stability and sensitivity for each genotype, and these ideas were extended here to develop FAST for either FCR tolerance or yield potential.

Recalling that FCR tolerance is the (yield) difference between the FCR-n and FCR-i *treatments* then it followed that the set of FCR tolerance effects for each environment and genotype was given by

$$\boldsymbol{u}_t = \boldsymbol{K}_t \boldsymbol{u}$$

where $\boldsymbol{u}_t$ is the $4m \times 1$ vector of FCR tolerance effects for each environment and genotype, and

$$\boldsymbol{K}_t = \boldsymbol{I}_4 \otimes \boldsymbol{K}_{t_2} \otimes \boldsymbol{I}_{36}$$

and $\boldsymbol{K}_{t_2} = \begin{bmatrix} -1 & 1 \end{bmatrix}$, where the levels of FCR *treatment* were ordered 1 = FCR-n and 2 = FCR-i.

Hence the separable latent regression model for $\boldsymbol{u}_t$ was

$$\boldsymbol{u}_t = (\boldsymbol{\Lambda}_s \otimes \boldsymbol{K}_{t_2} \otimes \boldsymbol{I}_{36})\boldsymbol{f}_s + \boldsymbol{K}_t \boldsymbol{\delta}_s$$

while the non-separable latent regression model for $\boldsymbol{u}_t$ was

$$\boldsymbol{u}_t = (\boldsymbol{\Lambda}_{tns} \otimes \boldsymbol{I}_{36})\boldsymbol{f}_{ns} + \boldsymbol{K}_t \boldsymbol{\delta}_{ns}$$

where $\boldsymbol{\Lambda}_{tns} = (\boldsymbol{I}_4 \otimes \boldsymbol{K}_{t_2})\boldsymbol{\Lambda}_{ns}$. We note that the incremental crop tolerance index introduced by Lemerle et al. [42] could be derived using the separable model by letting $\boldsymbol{K}_{t_2} = \begin{bmatrix} -\beta & 1 \end{bmatrix}$, where β was the common slope of the genetic regression of FCR-i on FCR-n across all environments. This simple measure could not be used for a non-separable variance model. It was then straightforward to apply the FAST approach to FCR tolerance obtaining overall performance (OP) measures for FCR tolerance index, stability of FCR tolerance and so on (see Smith and Cullis [41] for details).

Similarly defining yield potential as the yield in the absence of FCR then it followed that

$$\boldsymbol{u}_n = \boldsymbol{K}_n \boldsymbol{u}$$

where $u_n$ is the $4m \times 1$ vector of FCR-n effects and

$$K_n = I_4 \otimes K_{n_2} \otimes I_{36}$$

and $K_{n_2} = \begin{bmatrix} 1 & 0 \end{bmatrix}$. Therefore, the FAST approach could also be applied to yield potential.

## Results

The series of field experiments captured a range of environmental conditions (Fig 1), with drier than average conditions in 2015, high rainfall in 2016, drought conditions in 2017, and the effect of irrigation in 2018. The trial results captured the impact of environmental conditions on FCR disease pressure and this diversity of conditions provided a good test of heritable FCR tolerance in the durum genotypes. Higher levels of disease developed in the FCR inoculated plots and produced visually observable differences between FCR inoculated plots and the non-inoculated controls next to them. FCR inoculated plots were generally less vigorous with lower biomass and they took longer to reach ear emergence relative to the FCR treated plot (data not presented). These differences resulted in lower yield in the treated plots (Fig 3).

### Analysis of CRI

The fit of the CS and CORGH models were very similar for the two years (2016 and 2017). The CORGH model was chosen as this model avoided estimates of the variance component for the interaction Genotype:FCRTrt being fitted at zero and provided for the biologically sensible variance heterogeneity between the set of FCR-n and FCR-i effects. A summary of REML estimates of the genetic variance parameters for the analysis of CRI for 2016 and 2017 is presented in Table 4. In 2016, the variance components for CRI from FCR-i and FCR-n were small and were not statistically different from zero and the correlation between CRI for FCR-n and FCR-i was small and not significantly different from zero. This was consistent with the low FCR disease pressure in that year due to the high rainfall (Fig 1). In contrast, in 2017 the variance components for CRI from FCR-i and FCR-n were large, with the FCR-i being significantly different from zero. There was a strong correlation between CRI from FCR-n and FCR-i in 2017.

A scatter plot of the 2017 E-BLUPS of CRI for FCR-i against FCR-n (Fig 2) illustrates the rankings for resistance among genotypes with Suntop being the most resistant and TD1701 the most susceptible. DBA Bindaroi, together with three genotypes derived from crosses with 2–49, viz., V114926, V114916 and V114942, showed low CRI and hence good resistance but below the level of Suntop. EGA Bellaroi, which is generally considered the most susceptible genotype, performed similar to Hyperno for CRI. Several genotypes, including V101030 and TD1702, showed higher levels of CRI and hence higher susceptibility than EGA Bellaroi in this data.

Scatter plot of CRI for 2016 is included to show the low levels of FCR incidence in this season although the genetic variance parameters were not significant.

### Analysis of grain yield and FCR tolerance

There was strong evidence from the results of fitting the baseline model that a separable variance model would not be a sensible model for this data, with the correlation between the

**Table 4. Summary of REML estimates of the genetic variance parameters for the analysis of CRI for 2016 and 2017.**

| Parameter | 2016 Estimate | p-value | 2017 Estimate | p-value |
|---|---|---|---|---|
| var(FCR-n) | 0.1328 | 0.105 | 0.3596 | 0.083 |
| var(FCR-i) | 0.0996 | 0.154 | 0.6159 | 0.006 |
| corr(FCR-n, FCR-i) | -0.0665 | 0.978 | 0.5716 | 0.207 |

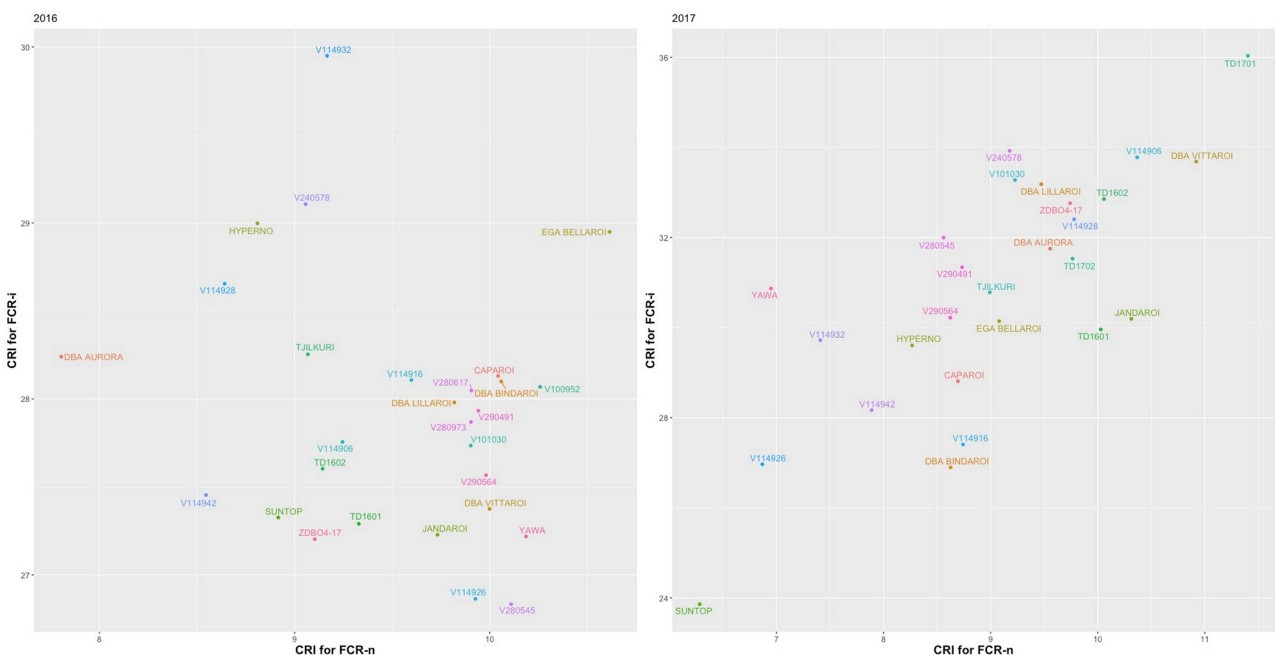

**Fig 2. Scatter plot of E-BLUPS of CRI for FCR-i against those for FCR-n for 2017.**

effects for FCR-i and FCR-n for each year varying from 0.19 to 0.88 (Table 5). Interestingly there was much agreement between the trials in 2015 and 2017, and those in 2016 and 2018. This was consistent with the seasonal conditions experienced in these years. The REML likelihood ratio statistic based on the baseline model for the presence of genetic variance associated with FCR tolerance for the four years was 53.525 on 4 df (p < 0.001).

In the next step, five variance models which incorporated correlation between environments were fitted to the variance structure of the VFE effects and these are summarised in Table 6. These variance models included two separable and three non-separable models. The AIC values (Table 6) indicated the superiority of non-separable models. This was not surprising given the variance and correlation heterogeneity which was observed from the fit of the baseline model. A FA(3) model provided the best fit among the non-separable models, again demonstrating the complexity of the variance structure of the VFE effects.

Table 7 presents the REML estimate of the variance matrix ($\mathbf{G_{ns}}$) for the concatenated factor EnvFCRTrt using a FA(3) non-separable model. Values on the upper triangle are the estimated correlations between the VFE effects for each level of EnvFCRTrt and those on the diagonals are the estimated variances of the VFE effects for each level of EnvFCRTrt. The values for the variances and correlations between the VFE effects for FCR-n and FCR-i within each

**Table 5. Summary of REML estimates of the genetic variance parameters for the analysis of grain yield using the baseline model.**

| Parameter | Estimate | | | |
|---|---|---|---|---|
| | 2015 | 2016 | 2017 | 2018 |
| var(FCR-n) | 0.0279 | 0.2845 | 0.1173 | 0.1523 |
| var(FCR-i) | 0.0481 | 0.2332 | 0.0370 | 0.1784 |
| cor(FCR-n, FCR-i) | 0.1939 | 0.8869 | 0.1300 | 0.8312 |

**Table 6. Summary of models fitted for grain yield from 2015–2018 trials: Number of genetic variance parameters, REML log-likelihood and the AIC values.**

| Type | Variance Model | nparm | logl | AIC |
|------|----------------|-------|------|-----|
| Separable | fa1(Env)×corgh-c(FCR) | 10 | 396.3 | -732.6 |
| Separable | corgh(Env)×corgh-c(FCR) | 12 | 406.2 | -746.4 |
| Non-separable | fa1(EnvFCRTrt) | 15 | 408.5 | -743.0 |
| Non-separable | fa2(EnvFCRTrt) | 23 | 417.8 | -747.5 |
| Non-separable | fa3(EnvFCRTrt) | 31 | 425.9 | -751.7 |

**Table 7. REML estimate of the variance matrix ($G_{ns}$) for the concatenated factor `EnvFCRTrt` using a FA(3) non-separable model.**

|  | 15-n | 15-i | 16-n | 16-i | 17-n | 17-i | 18-n | 18-i |
|--|------|------|------|------|------|------|------|------|
| **15-n** | 0.028 | 0.213 | 0.323 | 0.197 | 0.593 | -0.272 | 0.154 | -0.036 |
| **15-i** |  | 0.050 | 0.381 | 0.535 | 0.392 | 0.635 | -0.172 | 0.084 |
| **16-n** |  |  | 0.327 | 0.900 | 0.784 | 0.277 | -0.870 | -0.734 |
| **16-i** |  |  |  | 0.265 | 0.700 | 0.564 | -0.772 | -0.514 |
| **17-n** |  |  |  |  | 0.113 | 0.163 | -0.500 | -0.449 |
| **17-i** |  |  |  |  |  | 0.043 | -0.259 | 0.186 |
| **18-n** |  |  |  |  |  |  | 0.136 | 0.831 |
| **18-i** |  |  |  |  |  |  |  | 0.163 |

Values in the upper triangle are the estimated correlations between the VFE effects for each level of `EnvFCRTrt` and those on the diagonals are the estimated variances of the VFE effects for each level of `EnvFCRTrt`.

environment are almost identical to the values presented in Table 4 indicating the quality of the FA(3) fit.

Table 8 presents the REML estimates of the rotated loadings for FCR tolerance and yield for FCR-n respectively. Table 9 presents the REML estimate of the genetic correlation between environments for FCR tolerance, and between FCR tolerance and yield predictions for both FCR-n and FCR-I, for each pair of environments. There were moderate positive correlations between trials for FCR tolerance (Table 9 and Fig 4). The complexity of the genotype by environment interaction for FCR tolerance is less than that for FCR-n yield, which is reflected in the loadings for each of these two traits. Since the loadings for the first factor are all the same sign it is therefore possible to derive the overall performance for FCR tolerance for these environments.

Table 10 presents a summary of the FCR tolerance indices for each genotype, ordered on the FCR index along with accuracy (expressed as a percent) and 95% coverage intervals for the FCR index. These results provide clear evidence of genetic diversity in durum germplasm. In particular, conventional durum genotypes such as V101030, TD1702, V11TD013*3X-63 and DBA Bindaroi exhibit good FCR tolerance and were similar to the FCR tolerance of V114916

**Table 8. REML estimate of rotated loadings for FCR tolerance and yield for FCR-n and FCR-i.**

| Trial | FCR Tolerance | | | Yield for FCR-n | | | Yield for FCR-i | | |
|-------|-------|-------|-------|-------|-------|-------|-------|-------|-------|
|  | Load1 | Load2 | Load3 | Load1 | Load2 | Load3 | Load1 | Load2 | Load3 |
| 2015 | 0.168 | 0.107 | 0.031 | 0.049 | 0.158 | -0.033 | 0.103 | 0.129 | 0.030 |
| 2016 | 0.186 | -0.020 | -0.072 | 0.571 | 0.022 | 0.011 | 0.485 | 0.084 | 0.023 |
| 2017 | 0.310 | -0.091 | 0.034 | 0.258 | 0.134 | 0.010 | 0.099 | 0.167 | -0.075 |
| 2018 | 0.142 | 0.098 | -0.017 | -0.327 | 0.167 | 0.022 | -0.267 | 0.264 | 0.026 |

**Table 9. REML estimates of the genetic correlation between FCR tolerance and predicted yield for FCR-n for pairs of traits and trials.**

| Trial | | FCR Tolerance | | | | Yield FCR-i | | | | Yield FCR-n | | | |
|---|---|---|---|---|---|---|---|---|---|---|---|---|---|
| | | 2015 | 2016 | 2017 | 2018 | 2015 | 2016 | 2017 | 2018 | 2015 | 2016 | 2017 | 2018 |
| FCR Tolerance | 2015 | 1.000 | 0.433 | 0.476 | 0.598 | 0.750 | 0.345 | 0.753 | 0.100 | -0.486 | 0.122 | -0.050 | -0.258 |
| | 2016 | | 1.000 | 0.627 | 0.457 | 0.230 | 0.000 | 0.528 | 0.622 | -0.335 | -0.436 | -0.355 | 0.401 |
| | 2017 | | | 1.000 | 0.418 | 0.002 | -0.321 | 0.421 | 0.519 | -0.701 | -0.563 | -0.826 | 0.312 |
| | 2018 | | | | 1.000 | 0.430 | 0.339 | 0.753 | 0.430 | -0.315 | 0.106 | 0.013 | -0.144 |
| Yield FCR-i | 2015 | | | | | 1.000 | 0.535 | 0.636 | 0.084 | 0.213 | 0.381 | 0.393 | -0.172 |
| | 2016 | | | | | | 1.000 | 0.564 | -0.514 | 0.197 | 0.900 | 0.700 | -0.772 |
| | 2017 | | | | | | | 1.000 | 0.186 | -0.273 | 0.277 | 0.163 | -0.259 |
| | 2018 | | | | | | | | 1.000 | -0.036 | -0.734 | -0.449 | 0.831 |
| Yield FCR-n | 2015 | | | | | | | | | 1.000 | 0.323 | 0.593 | 0.154 |
| | 2016 | | | | | | | | | | 1.000 | 0.784 | -0.870 |
| | 2017 | | | | | | | | | | | 1.000 | -0.500 |
| | 2018 | | | | | | | | | | | | 1.000 |

and V114942 which originated from crosses with 2–49, a bread wheat genotype with the highest level of partial resistance to FCR. As expected, EGA Bellaroi had very low FCR tolerance and Suntop possessed the highest FCR tolerance index of all the genotypes. Yawa, a high yielding genotype, possessed the lowest FCR tolerance index.

Graphs of E-BLUPs of yield for FCR-i against FCR-n for each environment (Fig 3) highlight how the relationships vary between environments, most likely associated with differences in moisture conditions during the seasons. The 2016 experiment occurred in a high rainfall season and the 2018 trial was irrigated. Both 2015 and 2017 seasons were drought affected (Fig 1). The REML estimates of the correlations between the FCR-n and FCR-i effects for each genotype (Table 6) concurred with these plots, showing that the correlation was low in 2015 and 2017, but quite high for 2016 and 2018 respectively. This demonstrates the strong effect of environmental conditions on the expression of yield in the presence of FCR. Whilst there were differences between years for the correlations between FCR tolerance and both FCR-i and FCR-n (Fig 5 and Table 9), FCR tolerance was moderate to strongly negatively correlated with FCR-n. However, with FCR-i, there was a strong positive correlation in 2015 and a positive but moderate correlation in 2017 and 2018 seasons. In 2016, which was a high rainfall season, there was no correlation between FCR tolerance and FCR-i. These results indicated that selection based purely on experiments with only an inoculated FCR treatment could lead to tolerant selections.

## Discussion

Addition of FCR inoculum was very effective in producing a higher disease severity (CRI) and a higher percentage of *Fusarium* in the plates compared with the FCR-n control plots (data not presented). This, in turn, produced measurable differences in yield performance between FCR-i and FCR-n plots for all the varieties including the bread wheat control, Suntop. These differences were observed in all four seasons, including 2016, which resulted in low levels of CRI due to high rainfall in spring and favourable moisture conditions through the growing season (Figs 1 and 2). Similar observations were made Hollaway et al. [13].

There were significant differences in FCR development and the yield loss response of the genotypes between seasons. The relationship between yield from FCR-i plots and that from FCR-n plots was stronger in the wet seasons (2016 and 2018) than in the dry seasons (2015

**Table 10. Summary of the FCR tolerance indices for each genotype, ordered on the FCR index along with, accuracy (expressed as a percent) and 95% coverage intervals for the FCR index.**

| Genotype | OP (t/ha) | Accuracy | 95% Coverage Interval | |
|---|---|---|---|---|
| | | | Lower | Upper |
| SUNTOP | 0.309 | 88.9 | 0.128 | 0.490 |
| V114916 | 0.294 | 89.2 | 0.116 | 0.473 |
| V101030 | 0.270 | 89.1 | 0.091 | 0.449 |
| TD1702 | 0.239 | 80.5 | 0.004 | 0.473 |
| V114942 | 0.196 | 89.0 | 0.015 | 0.376 |
| V11TD013*3X-63 | 0.186 | 59.0 | -0.133 | 0.506 |
| DBA BINDAROI | 0.179 | 83.2 | -0.040 | 0.399 |
| V114908 | 0.126 | 69.0 | -0.160 | 0.412 |
| V280973 | 0.124 | 80.9 | -0.108 | 0.356 |
| V114928 | 0.114 | 89.1 | -0.066 | 0.293 |
| V281019 | 0.091 | 68.6 | -0.196 | 0.379 |
| V280545 | 0.048 | 87.2 | -0.146 | 0.241 |
| CAPAROI | 0.046 | 89.1 | -0.133 | 0.226 |
| JANDAROI | 0.040 | 89.0 | -0.140 | 0.220 |
| V100952 | 0.031 | 80.9 | -0.201 | 0.263 |
| V240578 | 0.017 | 87.1 | -0.177 | 0.212 |
| TD1701 | 0.015 | 80.4 | -0.220 | 0.250 |
| V280617 | 0.004 | 80.9 | -0.229 | 0.237 |
| V114926 | 0.003 | 86.8 | -0.193 | 0.200 |
| V114906 | 0.003 | 87.2 | -0.191 | 0.196 |
| TJILKURI | -0.026 | 80.8 | -0.259 | 0.207 |
| HYPERNO | -0.033 | 89.1 | -0.212 | 0.146 |
| DBA VITTAROI | -0.033 | 83.3 | -0.252 | 0.185 |
| ZDBO4-17 | -0.037 | 87.2 | -0.231 | 0.156 |
| TD1602 | -0.056 | 83.3 | -0.275 | 0.163 |
| DBA LILLAROI | -0.074 | 89.0 | -0.254 | 0.106 |
| 10TD033*3X-098 | -0.075 | 59.0 | -0.394 | 0.245 |
| V290564 | -0.105 | 87.0 | -0.300 | 0.090 |
| DBA AURORA | -0.149 | 89.0 | -0.330 | 0.031 |
| V290328 | -0.168 | 68.8 | -0.455 | 0.119 |
| TD1601 | -0.185 | 83.3 | -0.404 | 0.034 |
| V114932 | -0.194 | 87.2 | -0.388 | -0.000 |
| V290222 | -0.230 | 68.9 | -0.517 | 0.056 |
| EGA BELLAROI | -0.263 | 89.1 | -0.442 | -0.084 |
| V290491 | -0.284 | 87.2 | -0.477 | -0.090 |
| YAWA | -0.422 | 89.1 | -0.602 | -0.243 |

and 2017). This could most likely be due to the poor expression of yield potential in FCR-i plots in the dry seasons due to increased CRI. This observation was consistent with Hollaway et al. [13] who reported higher yield losses in seasons where rainfall in September/October period was below long-term average for the site. Despite these seasonal effects there was a consistent trend in the data for particular genotypes showing less than expected yield loss and thus a moderate to high correlation between FCR tolerance values between years (Fig 4). The two control varieties performed as expected with EGA Bellaroi showing the third lowest tolerance and Suntop showing the highest tolerance.

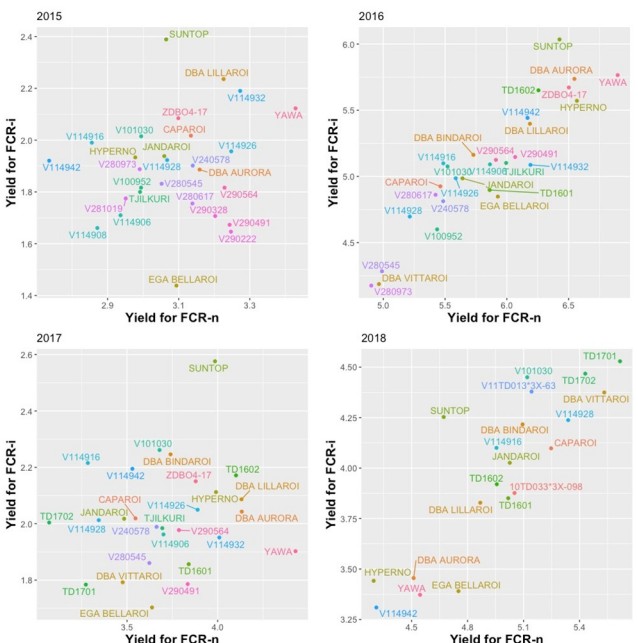

**Fig 3. Scatter plots of E-BLUPs of yield for FCR-i against FCR-n for each trial.**

In this study, there was a consistent and reasonably strong genotype effect showing low yield loss in certain genotypes. It was therefore considered better to use a simpler measure of tolerance that directly relates to the ability of a genotype to tolerate the disease and to produce a relatively higher yield rather than percentage yield loss or the regression method. This

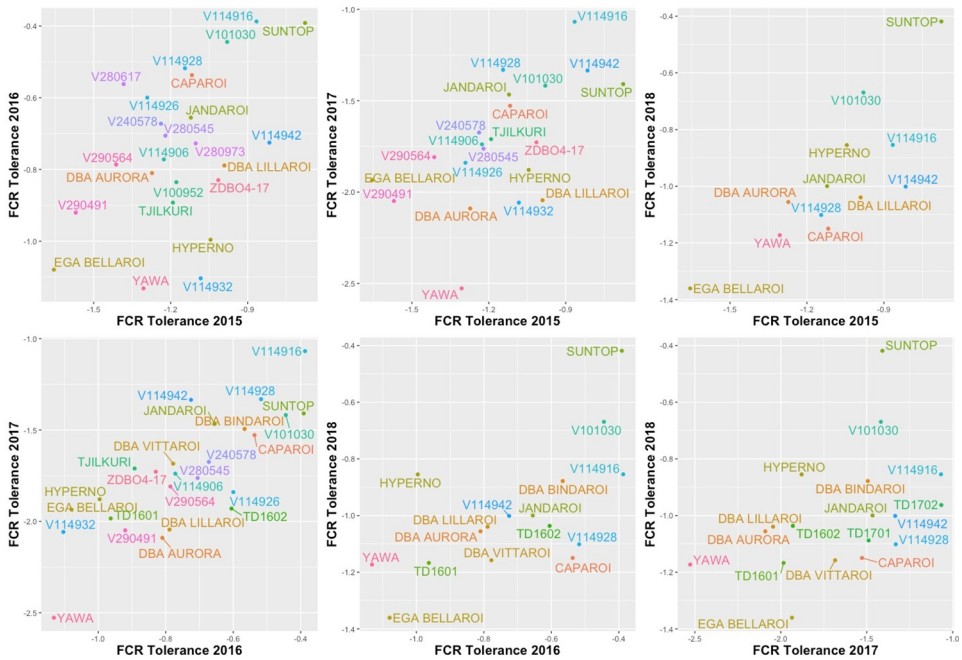

**Fig 4. Scatter plot of E-BLUPS of FCR tolerance for each pair of trials for those genotypes present in both years.**

method of using the simple difference between yield of FCR-n and FCR-i yields as the measure of FCR tolerance of genotypes using EBLUPs from a robust MET analysis has provided an objective method of determining the tolerance status of durum genotypes. However, some authors have used percentage yield loss [13, 19] and others have used a regression approach to achieve independence from yield potential of the genotypes [20, 29].

We analysed CRI as a separate trait and did not include it as a covariate in the analysis of yield because a univariate analysis of this trait showed it to be genetically driven in both the 2016 and 2017 trials and a key condition for the use of a covariate is that it cannot be affected by the treatment applied [43]. There was little difference between the genotypes within FCR treatments for CRI in 2016, possibly due to better growing conditions which limits disease expression. Also, there was little correlation between CRI from FCR-i and FCR-n plots in 2016. The range of CRI increased in the 2017 season (Fig 2), most likely due to dry conditions, and there was a strong correlation between CRI from FCR-i and FCR-n plots.

There were many examples of mismatches between CRI and FCR tolerance in this study. CRI of EGA Bellaroi for FCR-i treatment was lower than that of TD1702 and V101030 both of which ranked very high for FCR tolerance relative to EGA Bellaroi. V101030 showed relatively high CRI but it was consistently rated as highly tolerant (Fig 2 and Table 10). Likewise, there are mismatches between resistance ratings of genotypes in the Australian ACAS/NVT system (https://www.grdc-nvt.com.au/) and their performance for tolerance as shown in Table 10. EGA Bellaroi, Caparoi and Jandaroi are all rated very susceptible (VS) in the ACAS/NVT system but only EGA Bellaroi has been judged very intolerant in this study. Jandaroi and Caparoi tended to rank substantially better than EGA Bellaroi. DBA Bindaroi is rated susceptible–very susceptible (SVS) in the ACAS/NVT system but it was a better tolerant genotype in this study. Similar observations were made in a report on bread wheat FCR tolerance by Davies et al [29] regarding EGA Wylie, which is rated highly for resistance to FCR, but it showed very low level of FCR tolerance relative to Suntop and Spitfire. These examples demonstrate the difference between resistance and tolerance. Whilst their symptom development is similar to the susceptible genotypes, the tolerant genotypes are able to withstand the disease burden, keep functioning, set and fill more grains.

Whilst the exact mechanism of tolerance is not known, there is some circumstantial evidence that Caparoi tolerates terminal drought better than EGA Bellaroi. DBA Bindaroi is a further improvement over Caparoi in its tolerance of moisture stress (unpublished data). It is likely that tolerance, as measured in this study, is able to include benefits from all other traits, such as optimum maturity, better root architecture, stay green, leaf rolling and overall drought tolerance which are likely to be important in reducing the impact of FCR [1]. Thus, tolerance appears to be a superior and more comprehensive criterion than FCR resistance for assigning FCR ratings to genotypes as well as in breeding and selection.

This study is the first systematic and targeted investigation of durum germplasm for FCR tolerance. The previous studies were targeting FCR resistance and they reported a lack of genetic variation in durum and other tetraploid wheat species. Also, these previous studies evaluated FCR resistance either in an outdoor pot assay [27] or in the glasshouse [30] with an unspecified relationship to performance under field conditions. Moreover, genetic analysis of FCR resistance in bread wheat in crosses with 2–49, W21MMT70 and Mendos [44, 45] have shown occurrence of major QTLs for FCR resistance on D chromosomes which are not present in durum wheat. Another study reported transfer of a major FCR resistance locus on 3BL from CSCR6 (*Triticum spelta*) into durum but found that the transferred QTL was not effective in the durum background [30]. These results provided additional evidence for the lack of variation for FCR resistance in durum and this led to pre-breeding work to introgress

resistance genes from bread wheat [28] without consideration being given to the possibility of genetic variation for tolerance to FCR.

The concept of tolerance is relatively new in FCR research, first demonstrated by [17] in bread wheat but is well known in other pathosystems, such as, blackleg disease of canola [46, 47]. Whilst we have not investigated the genetic or physiological basis of FCR tolerance, it appears most likely a polygenic trait, similar to horizontal resistance to blackleg disease of canola. Similar to reports of effects of other plant traits on impact of FCR on yield, horizontal resistance to blackleg also appears to be a tolerance trait and it is confounded by other plant traits, such as flowering time, plant height and maturity [48]. Development of horizontal resistance to blackleg by repeated selection under heavy disease pressure (e.g. in disease nurseries in Lake Bolac, Australia) has been a major achievement in Australian canola breeding and it is the basis of the success of the canola industry to date [47, 48]. It could, likewise, be possible to develop FCR tolerant durum varieties by repeated selection under high FCR disease pressure using the methods described in this study.

The results for genotypes from crosses with 2–49 were generally similar to those for conventional durum genotypes with one 2–49 cross, V114916, performing similar to V101030, a conventional durum genotype with the best tolerance. Whilst there is evidence for these genotypes to have lost the D chromosomes quickly under pedigree selection [49] further work is needed to confirm chromosome numbers of this material. Evaluation of V114942 included in this study has shown markers for all seven chromosomes of the D genome in this genotype but V114916 (also developed from crossing with 2–49) did not appear to possess for any of the D chromosome markers (D Mather and A Binney, personal communication).

Tolerance to FCR demonstrated in durum wheat in this paper also occurs in bread wheat [17] and the correlation between FCR resistance and FCR tolerance was low. FCR tolerance of Caparoi, assessed in this study as the rate of change in yield, was comparable to that of Suntop although Caparoi showed higher FCR severity (increased susceptibility). Also, Sunguard which showed the lowest FCR severity (i.e., the highest resistance) possessed low FCR tolerance.

In this study FCR tolerance was strongly negatively correlated with yield predictions for FCR-n treatments in 2017, and moderately negatively related in 2015 and 2016 seasons (Fig 5). There were also significantly positive correlations observed between FCR tolerance values and yield from FCR-n treatments across the years (Table 9) and thus the overall correlation was moderately negative suggesting that it could be somewhat difficult to combine high yield potential under low disease or disease-free conditions and FCR tolerance. There are some examples, such as resistance from the *mlo* gene to barley powdery mildew (*Blumeria graminis* [syn. *Erysiphe graminis*] f. sp. *Hordei*) [50] and durable resistance to wheat leaf rust from the *Lr34* gene are associated with a yield penalty [51]. In a recent publication [52], the findings of the yield penalty have been revised and the authors have noted that *LR34* together with other durable resistance genes forms the backbone of the CIMMYT bread wheat germplasm. However, in FCR, tolerance is even less likely to result in yield penalty because, as commented above, tolerance could be considered to result from drought tolerance traits, such as, optimum maturity, better root architecture, stay green, leaf rolling etc. which are all yield positive traits. Good FCR tolerance is required for high yields under Australian conditions because a large portion of cereal crops surveyed in recent years are reported to contain significant levels of FCR infection [3]. Also, one of the genotypes identified as relatively tolerant in this study, DBA Bindaroi, has been a good performer for grain yield and grain quality in the Australian ACAS/NVT trials (https://www.grdc-nvt.com.au/) and it was released for commercial cultivation in 2017.

In conclusion, this study has provided conclusive evidence for the occurrence of significant variation within durum germplasm for FCR tolerance. FCR tolerance is a different trait to

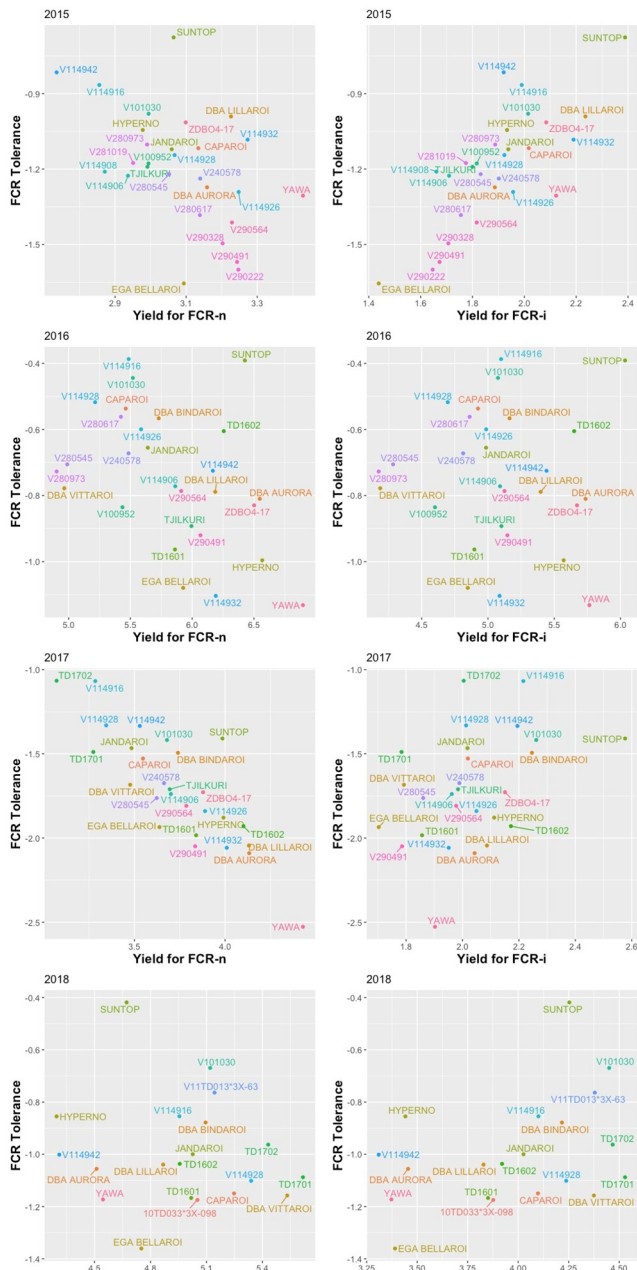

**Fig 5. Scatter plot of E-BLUPS of FCR tolerance against yield for FCR-n and FCR-i for each trial.**

FCR resistance, and it needs a different screening approach for selection in breeding programs compared with the conventional approach of screening based on CRI assessments. Considering the positive genetic correlation between FCR tolerance and EBLUPs from FCR-i treatment (Fig 5, Table 9) it could be possible to select indirectly for FCR tolerance based purely on experiments with only an inoculated FCR *treatment* as the first step and then progressing the best genotypes to properly designed tolerance trials with FCR-i and FCR-n plots to assess tolerance. This approach has been successful in the DBA Northern program as evidenced by genotypes such as TD1702 and V11TD013*3X-63 in this study. Future directions for

developing better tolerance in durum would be to focus on enriching the germplasm for drought tolerance traits which could have pleiotropic effects on FCR tolerance. Also, commencing preliminary FCR tolerance evaluation with FCR-i plots in early stages (e.g. Stage 2, i.e., the first year of replicated yield trials) in the breeding cycle would potentially avoid any loss of promising FCR-tolerant genotypes that may not meet other selection criteria.

## Supporting information

**S1 File.**
(DOCX)

## Acknowledgments

We thank Drs. Ky Matthews, Beverly Gogel and Alison Smith (UoW) for constructive discussions, Prof. Diane Mather and Allan Binney (The University of Adelaide) for D chromosome marker data for V114942, and the DBA breeding team and the cereal pathology team (NSW DPI) for their excellent technical assistance.

## Author Contributions

**Conceptualization:** Gururaj Pralhad Kadkol, Steven Simpfendorfer, Brian Cullis.

**Data curation:** Gururaj Pralhad Kadkol, Jess Meza, Steve Harden.

**Formal analysis:** Jess Meza, Steve Harden, Brian Cullis.

**Funding acquisition:** Gururaj Pralhad Kadkol.

**Investigation:** Gururaj Pralhad Kadkol, Steven Simpfendorfer.

**Methodology:** Gururaj Pralhad Kadkol, Steven Simpfendorfer, Steve Harden, Brian Cullis.

**Project administration:** Gururaj Pralhad Kadkol, Brian Cullis.

**Resources:** Gururaj Pralhad Kadkol, Steven Simpfendorfer.

**Supervision:** Gururaj Pralhad Kadkol, Steven Simpfendorfer, Brian Cullis.

**Validation:** Gururaj Pralhad Kadkol, Steven Simpfendorfer.

**Visualization:** Jess Meza.

**Writing – original draft:** Gururaj Pralhad Kadkol.

**Writing – review & editing:** Gururaj Pralhad Kadkol, Jess Meza, Steven Simpfendorfer, Steve Harden, Brian Cullis.

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
