## [Decision Letter · Decision Letter 0]

6 Nov 2020

PONE-D-20-30596

Genetic Variance for Fusarium Crown Rot Tolerance in Durum Wheat

PLOS ONE

Dear Dr. Kadkol,

Thank you for submitting your manuscript to PLOS ONE. After careful consideration, we feel that it has merit but does not fully meet PLOS ONE’s publication criteria as it currently stands. Therefore, we invite you to submit a revised version of the manuscript that addresses the points raised during the review process.

Authors are invited to modify the text according to reviewer's suggestions, in order to render the paper acceptable for publicaton.

We look forward to receiving your revised manuscript.

Kind regards,

Sabrina Sarrocco

Academic Editor

PLOS ONE

Journal Requirements:

Reviewers' comments:

Reviewer's Responses to Questions

**Comments to the Author**

1. Is the manuscript technically sound, and do the data support the conclusions?

Reviewer #1: Yes

2. Has the statistical analysis been performed appropriately and rigorously? 

Reviewer #1: Yes

3. Have the authors made all data underlying the findings in their manuscript fully available?

Reviewer #1: Yes

4. Is the manuscript presented in an intelligible fashion and written in standard English?

Reviewer #1: Yes

5. Review Comments to the Author

Reviewer #1: Dear corresponding author,

The manuscript number PONE-D-20-09326 titled “Genetic variance for Fusarium crown rot tolerance in durum wheat” investigate the tolerance to the cereal disease Fusarium crown rot (FCR) in a set of 34 durum wheat genotypes in a series of replicated field trials over four years with inoculated and non inoculated plots of the genotypes. The nature of the subject studied and the results obtained are worthy to be taken into account for publication in PLOS ONE. This paper is well written and well organized. However, there are some minor points that need to be clarify and a revision of the manuscript could be help to increase the article quality as well as its impact on the scientific community. For this reason, I provided a revision list. I hope that my considerations could be useful to improve your study and its clarity to reader.

Revision list:

1) Line 16: Please change “checks” with “controls”;

2) Line 39: Please be more precise and avoid the word “overseas”;

3) Line 41: Please indicate what mean “NSW”;

4) Line 49: The author mention that there are no effective fungicide products available for management of FCR but maybe should be mentioned seed dressing;

5) Lines 125-126: What is meant for “current elite Australian material”? Please specify;

6) Lines 126-128: This sentence is not clear;

7) Lines 125-131: To my opinion insert in the aim of the research, other additional comment could be not useful for the reader. I suggest to reformulate all these sentences;

8) Line 139-141: Please avoid the general use of the word “preliminary”, I suggest to be more precise in describe these preliminary trials;

9) Line 156 and lines 164-165: You don’t have the same genotypes each year. Some genotypes were tested only for one year and some other for two years and some others again for three years. Additional promising lines were added every year. Please describe better the reasons of this choice;

10) Line 172: What type of isolates did you use? Please be more precise about name, origin, host from which were obtained, etc.;

11) Lines 169-174: Please describe better the inoculation technique on experimental plots, provide more details;

12) Line 542: DISCUSSION SECTION: I suggest to revise the entire section taking care to expand the results of this study in a wider research context. In the present form there are many indication about this study but a low level of citations of other studies that could incorporate this research in a more comprehensive research scenario;

13) Lines 547-549: This sentence is not clear, please reformulate it;

14) Lines 568-569: I suggest to avoid the use of this sentence;

15) Lines 611-612: The authors mention an uncertain relationship of the previous studies with field conditions, please be more accurate about this statement;

16) Lines 622-624: This assertion is of crucial importance. Please expand it possibly adding some other studies (with references) were the difficult to combine high yield potential and disease tolerance were found;

17) Lines 626-629: This sentence could be important but I think they need of a reference;

18) Lines 631-632: What pathosystem? Please be more accurate;

Best regards.

6. PLOS authors have the option to publish the peer review history of their article (what does this mean?). If published, this will include your full peer review and any attached files.

Reviewer #1: No

---

## [Author Response · Author response to Decision Letter 0]

22 Dec 2020

Response to Reviewer

PONE-D-20-09326 - “Genetic variance for Fusarium crown rot tolerance in durum wheat”

We thank the reviewer for his kind comments on our paper and helpful suggestions. We have accepted most of their suggestions and provided valid reasons where we didn’t accept their suggestions.

1. Line 16: Please change “checks” with “controls”;

• Agreed. “checks” changed to “controls”

2. Line 39: Please be more precise and avoid the word “overseas”;

• Agreed. The sentence has been changed to “…..is an important disease of cereals in Australia and other countries and regions, such as, USA, South Africa, North Africa, Italy, Middle East and China (Burgess et al. 2001; Smiley et al. 2005; Alahmad et al, 2018, Simpfendorfer et al. 2019).”

3. Line 41: Please indicate what mean “NSW”;

• Sentence changed to “…………..production in northern New South Wales (NSW) and southern Queensland……………..”

4. Line 49: The author mention that there are no effective fungicide products available for management of FCR but maybe should be mentioned seed dressing;

• We have amended the original sentence to “Currently, there are no effective seed or foliar fungicide products available for the management of FCR (Alahmad et al. 2018)”. 

• As discussed by Alahmad et al. (2018) the various chemical treatments tried either as post-emergent sprays or as seed treatments have shown limited or no efficacy for FCR control under field conditions. Currently, registered seed treatments in Australia are only registered for suppression of Fusarium crown rot which have limited activity. Seed treatments are really only for limiting seedling blight when sowing Fusarium head blight infected grain. This does not provide significant protection from infection coming from infected stubble during the season, which is the main pathway with Fusarium crown rot infection.

5. Lines 125-126: What is meant for “current elite Australian material”? Please specify;

• Sentence amended to “…………..current elite Australian durum breeding material in the DBA program.”

6. Lines 126-128: This sentence is not clear;

• See below

7. Lines 125-131: To my opinion insert in the aim of the research, other additional comment could be not useful for the reader. I suggest to reformulate all these sentences;

• Sentences amended to “Previous studies of disease tolerance have used several levels of disease pressure to determine tolerance in intensive tests of a small number of genotypes (Raberg et al. 2009; Forknall et al. 2019). However, this approach is not suitable for estimating genetic variation in a breeding setting because a higher number of genotypes need to be tested without making the experiments too large and cost prohibitive”.

8) Line 139-141: Please avoid the general use of the word “preliminary”, I suggest to be more precise in describe these preliminary trials;

• We would like to retain the original sentence which describes the nature of the work conducted prior to 2015. These trials were used to determine an optimum protocol for the 2015-17 trials that are presented in the manuscript and they were preliminary in nature.

9) Line 156 and lines 164-165: You don’t have the same genotypes each year. Some genotypes were tested only for one year and some other for two years and some others again for three years. Additional promising lines were added every year. Please describe better the reasons of this choice;

• Our reasoning was that it should be possible to screen for FCR tolerance using experiments with only an inoculated FCR treatment. We have demonstrated this by adding putatively better tolerant lines from other FCR treated trials into this study. We have discussed this in the Discussion section (lines 667-671).

• The text has been amended to “Some lines judged to be FCR-intolerant were replaced with new lines that were selected based on their performance in other FCR treated trials in the DBA North breeding program to improve the chances of identifying FCR-tolerant lines. However, a fairly high degree of concurrence between trials was maintained and this allowed the MET analysis of the data as discussed below”.

10) Line 172: What type of isolates did you use? Please be more precise about name, origin, host from which were obtained, etc.;

• We have added the following text and table: “ Although FCR isolates have been shown to vary in aggressiveness, there is no race structure established that causes differential varietal reactions (Akinsanmi et al. 2004). The details of the five isolates has been added as below: All isolates were established from hyphal tip cultures to ensure purity and identified by qPCR to be Fusarium pseudograminearum by SARDI. Isolates were collected from different commercial wheat crops with severe basal browning characteristic of Fusarium crown rot infection at harvest in either 2013 or 2014”.

• Table 3 Details of Fp isolates used in preparation of CR inoculum 

SARDI ID Isolate ID Year State Location Host qPCR ID

4093 AC29312

 CAS-13/94C

 2013

 NSW Walgett Wheat F. pseudograminearum

4093 AC29363

 CAS-13/131C

 2013 NSW Rowena Wheat F. pseudograminearum

4095 AC29475

 CAS-13/161N

 2013 NSW Moree Durum F. pseudograminearum

4849 BA48166

 CAS-14/98C

 2014 NSW Warren Wheat F. pseudograminearum

4719 BA40558

 CAS 14/88N

 2014 Qld Moonie Wheat F. pseudograminearum

11) Lines 169-174: Please describe better the inoculation technique on experimental plots, provide more details;

• Inoculation method is described in lines 152-153. “Plots inoculated with FCR were sown with inoculum mixed with viable seed at a rate of 2 g Fp inoculum/m row, as described by Dodman and Wildermuth (1987).” Full details of inoculum production and inoculation are outlined in Forknall et al. 2019

12) Line 542: DISCUSSION SECTION: I suggest to revise the entire section taking care to expand the results of this study in a wider research context. In the present form there are many indication about this study but a low level of citations of other studies that could incorporate this research in a more comprehensive research scenario;

• We wish to retain most of the original text but have incorporated the reviewer’s specific suggestions (13-15) because:

As we have explained, FCR tolerance is a new concept and there is little work reported in this area of study as reviewed by Forknall et al. (2019).

There have been no published reports of genetic variance in FCR tolerance in either bread wheat or durum wheat (except for reported tolerance in Caparoi by Forknall et al., 2019).

This study is the first demonstration of variation for FCR tolerance in durum wheat.

13) Lines 547-549: This sentence is not clear, please reformulate it;

• The sentence has been amended to “These differences were observed in all four seasons, including 2016, which resulted in low levels of CRI due to high rainfall in spring and favourable moisture conditions through the growing season (Figs 1 and 2) and this was consistent with the observations of Hollaway et al. (2013)”.

• Also, an additional graph of 2016 CRI data has been added to Fig 2 to support the comments.

14) Lines 568-569: I suggest to avoid the use of this sentence;

• Agreed. Sentence deleted.

15) Lines 611-612: The authors mention an uncertain relationship of the previous studies with field conditions, please be more accurate about this statement;

• “uncertain” amended to “unspecified”

16) Lines 622-624: This assertion is of crucial importance. Please expand it possibly adding some other studies (with references) were the difficult to combine high yield potential and disease tolerance were found;

• Thanks for this comment. We have amended the paragraph and added discussion on the likelihood of FCR tolerance have a yield penalty. We have also moved this paragraph towards the end of the discussion for better logical flow.

• Text has amended to “In this study FCR tolerance was strongly negatively correlated with yield predictions for FCR-n treatments in 2017, and moderately negatively related in 2015 and 2016 seasons (Fig 5). There were also significantly positive correlations observed between FCR tolerance values and yield from FCR-n treatments across the years (Table 9) and thus the overall correlation was moderately negative suggesting that it could be somewhat difficult to combine high yield potential under low disease or disease-free conditions and FCR tolerance. There are some examples, such as resistance from the mlo gene to barley powdery mildew (Blumeria graminis [syn. Erysiphe graminis] f. sp. Hordei) (Brown and Rant, 2013) and durable resistance to wheat leaf rust from the Lr34 gene are associated with a yield penalty (Singh and Huerta-Espino, 1997). In a recent publication (Huerta-Espino et al. 2020), the findings of the yield penalty have been revised and the authors have noted that LR34 together with other durable resistance genes forms the backbone of the CIMMYT bread wheat germplasm. However, in FCR, tolerance is even less likely to result in yield penalty because, as commented above, tolerance could be considered to result from drought tolerance traits, such as, optimum maturity, better root architecture, stay green, leaf rolling etc. which are all yield positive traits. Good FCR tolerance is required for high yields under Australian conditions because a large portion of cereal crops surveyed in recent years are reported to contain significant levels of FCR infection (Simpfendorfer et. al. 2019). Also, one of the genotypes identified as relatively tolerant in this study, DBA Bindaroi, has been a good performer for grain yield and grain quality in the Australian ACAS/NVT trials (https://www.grdc-nvt.com.au/) and it was released for commercial cultivation in 2017..”

17) Lines 626-629: This sentence could be important but I think they need of a reference;

• Reference (website address) added.

18) Lines 631-632: What pathosystem? Please be more accurate;

• Sentence amended to “The concept of tolerance is relatively new in FCR research, first demonstrated by Forknall et al. (2019) in bread wheat but is well known in other pathosystems, such as, blackleg disease of canola (Salisbury et al. 1995; Raman et al. 2013).”

We have also taken this opportunity to replace “line” with “genotype” for better consistency.

We have also made other corrections including a small change to the title of the paper. We trust this is okay.

We thank the referee for his time and constructive comments.

Gururaj Kadkol

(on behalf of all co-authors)

---

## [Decision Letter · Decision Letter 1]

13 Jan 2021

Genetic Variation for Fusarium Crown Rot Tolerance in Durum Wheat

PONE-D-20-30596R1

Dear Dr. Kadkol,

We’re pleased to inform you that your manuscript has been judged scientifically suitable for publication and will be formally accepted for publication once it meets all outstanding technical requirements.

Kind regards,

Sabrina Sarrocco

Academic Editor

PLOS ONE

Additional Editor Comments (optional):

Reviewers' comments:

Reviewer's Responses to Questions

**Comments to the Author**

1. If the authors have adequately addressed your comments raised in a previous round of review and you feel that this manuscript is now acceptable for publication, you may indicate that here to bypass the “Comments to the Author” section, enter your conflict of interest statement in the “Confidential to Editor” section, and submit your "Accept" recommendation.

Reviewer #1: (No Response)

2. Is the manuscript technically sound, and do the data support the conclusions?

Reviewer #1: (No Response)

3. Has the statistical analysis been performed appropriately and rigorously? 

Reviewer #1: (No Response)

4. Have the authors made all data underlying the findings in their manuscript fully available?

Reviewer #1: (No Response)

5. Is the manuscript presented in an intelligible fashion and written in standard English?

Reviewer #1: (No Response)

6. Review Comments to the Author

Reviewer #1: (No Response)

7. PLOS authors have the option to publish the peer review history of their article (what does this mean?). If published, this will include your full peer review and any attached files.

Reviewer #1: No

---

## [Editor Report · Acceptance letter]

25 Jan 2021

PONE-D-20-30596R1 

Genetic Variation for Fusarium Crown Rot Tolerance in Durum Wheat 

Dear Dr. Kadkol:

I'm pleased to inform you that your manuscript has been deemed suitable for publication in PLOS ONE. Congratulations! Your manuscript is now with our production department. 

Kind regards, 

on behalf of

Dr Sabrina Sarrocco 

Academic Editor

PLOS ONE